# Level of Serum Fetuin-A Correlates with Heart Rate in Obstructive Sleep Apnea Patients without Metabolic and Cardiovascular Comorbidities

**DOI:** 10.3390/ijerph19116422

**Published:** 2022-05-25

**Authors:** Elżbieta Reichert, Jerzy Mosiewicz, Wojciech Myśliński, Andrzej Jaroszyński, Agata Stanek, Klaudia Brożyna-Tkaczyk, Barbara Madejska-Mosiewicz

**Affiliations:** 1Individual Specialist Medical Practice, Włostowicka 293 C St., 24-100 Puławy, Poland; elzbart@wp.pl; 2Department of Internal Diseases, Medical University of Lublin, Staszica 16 St., 20-081 Lublin, Poland; jerzy.mosiewicz@umlub.pl (J.M.); wojciechmyslinski@umlub.pl (W.M.); 3Collegium Medicum, Jan Kochanowski University in Kielce, 23-517 Kielce, Poland; jaroszynskiaj@interia.pl; 4Department and Clinic of Internal Medicine, Angiology and Physical Medicine, Faculty of Medical Sciences in Zabrze, Medical University of Silesia, Batorego 15 St., 41-902 Bytom, Poland; 5Department of Internal Diseases, 1st Public University Hospital No. 1 in Lublin, Staszica 16 St., 20-081 Lublin, Poland; klaudiabrozyna19@gmail.com (K.B.-T.); bmosiewicz@gmail.com (B.M.-M.)

**Keywords:** obstructive sleep apnea, Fetuin-A, polygraphy, cardiovascular diseases

## Abstract

Obstructive sleep apnea (OSA) is the most common type of sleep-induced breathing disorder in the adult population and significantly affects the condition of the cardiovascular system. Fetuin-A (Fet-A) is a hepato- and adipokine, which prevents vessel calcification, and its level correlates with atherogenesis and metabolic disorders. The associations of cardiovascular diseases (CVD) both with OSA, which increases CVD risk, and Fet-A, which prevents CVD, justify the question of their mutual interactions in patients with OSA. Therefore, we sought to analyze Fet-A as an early biomarker of CVD risk in OSA patients without metabolic and cardiovascular comorbidities except for properly controlled arterial hypertension. We have found that in these patients, OSA does not appear to directly affect Fet-A levels. However, high Fet-A levels were more common in the group of patients with OSA, and the hypopnea index was significantly higher among subjects with the highest Fet-A levels. The level of Fet-A in OSA patients positively correlates with pulse rate, and it does not correlate with pulse pressure in this group unlike in the control group, where such a relationship exists. To our best knowledge, this is the first study to analyze this relationship in OSA patients without any significant cardiovascular comorbidities.

## 1. Introduction

The term obstructive sleep apnea (OSA) describes episodes of upper airway obstruction during sleep lasting at least 10 s with total airway obstruction and thoracoabdominal effort, leading to a decrease in arterial blood oxygenation [1]. OSA is the most common type of sleep-induced breathing disorder and is quite often found in the adult population. In addition to the daytime and nighttime symptoms it causes, OSA is also associated with cardiovascular disease (CVD). In patients with OSA, the risk of hypertension [2], treatment resistance [3], ischemic heart disease [4], cardiovascular events, arrhythmias, heart failure [2], stroke [5], neurodegenerative diseases, and poor cognitive performance [6] is increased. One of the agents known to be involved in atherogenesis is Fetuin-A (Fet-A) [7,8]. Produced in the liver, Fet-A is a glycoprotein that limits the calcification of the vessels [9]. In conditions of low serum Fet-A concentration, the risk of cardiovascular events in patients with chronic kidney disease increases due to the potential development of coronary calcification [10]. It is associated with an increase in the incidence of sudden cardiac death, myocardial infarction, left ventricular hypertrophy, and heart failure [10,11]. Patients with obstructive sleep apnea (OSA) are at increased risk for cardiovascular diseases (CVDs). Fet-A has been associated with metabolic syndrome, insulin resistance, and type 2 diabetes mellitus, all of which are highly prevalent in patients with OSA and associated with increased CVD risk [12]. Conversely, high serum Fet-A levels prevent calcification of the mitral valve in patients with coronary artery disease [13]. Fet-A also inhibits calcification of the aortic valve in patients without coexisting diabetes and chronic kidney disease [14,15]. Fet-A concentration is also correlated with some metabolic disorders. It is higher in cases of reduced insulin sensitivity, in people with metabolic syndrome, and with atherogenic lipid profiles [16,17]. High levels of Fet-A have also been found in obese patients and patients with non-alcoholic fatty liver disease [18,19]. Considering the properties of Fet-A and its potential role in the pathophysiology of cardiovascular diseases, it may also become a useful marker in evaluating OSA patients. Therefore, in this study, we sought to analyze Fet-A as an early biomarker of CVD risk in OSA patients without metabolic and cardiovascular comorbidities except for properly controlled arterial hypertension.

## 2. Materials and Methods

### 2.1. Participants

The study was planned in a group of patients with suspected OSA (*n* = 192) aged 35–65 years without comorbidities except for properly controlled arterial hypertension. Based on the polygraphic (PG) results, patients were included in the study group when the PG result allowed for the diagnosis of OSA or in the control group when OSA was excluded based on PG. Out of 192 people suspected of OSA, 85 subjects were qualified for the study. Among them, 58 patients with newly diagnosed OSA were included in the study group. The control group consisted of 27 people with OSA excluded based on PG. The study group was dominated by men (75.9%), but their percentage did not differ significantly from the control group (66.7%). The presence of comorbidities was assessed with an interview obtained from the patient and an analysis of the available medical documentation. Each patient underwent a detailed, targeted medical interview assessing the risk of OSA using the Berlin Questionnaire and the severity of daytime somnolence according to the Epworth Sleepiness Scale [20,21]. Qualified subjects were informed of the study objectives and expressed their informed consent to participate. The study was approved by the Bioethics Committee at the Medical University of Lublin.

### 2.2. Body Height, Body Weight, Body Mass Index, Systolic and Diastolic Blood Pressure, Pulse Pressure, and Mean Arterial Pressure Measurements

Body height (BH) and body weight (BW) were measured with bare feet and light clothing. Body mass index (BMI) was calculated as BW (kg)/BH-squared (m^2^). Systolic blood pressure (SBP) is defined as the maximum pressure experienced in the aorta when the heart contracts and ejects blood into the aorta from the left ventricle. Diastolic blood pressure (DBP) is the minimum pressure in the aorta when the heart is relaxing before ejecting blood into the aorta from the left ventricle [22]. Blood pressure was measured noninvasively twice a day during the 2-day stay at the Sleep Laboratory. Systemic hypertension was considered as effectively treated when all four measured values of systolic blood pressure were <140 mm Hg and <90 mm Hg for diastolic blood pressure. Four measurements of systolic and diastolic blood pressure values were averaged. Pulse pressure (PP) is the difference between systolic (SBP) and diastolic (DBP) blood pressure [22] and was calculated as:PP = SBP − DBP

Mean arterial pressure (MAP) was calculated according to the following formula [23]:MAP = DBP + 1/3(SBP − DBP)

### 2.3. Polygraphic Examination

Each subject had PG performed. The diagnosis of breathing disorders during sleep was made by the overnight, unsupervised, simplified polygraphy using the devices SleepDoc Porti 6 and Porti 8 (Dr. Fenyves und Gut Deutschland GmbH). The following parameters were analyzed in the course of PG: respiratory movements of the chest and abdomen, airflow through the nose, blood saturation and pulse rate (percutaneous pulse oximetry), body position, and snoring. The following parameters were analyzed:Apnea–hypopnea index (AHI)—the number of apnea and hypopnea episodes per hour of the examination;Apnea index (AI)—the number of apnea episodes per hour of examination;Hypopnea index (HI)—number of hypopnea episodes per hour of the test;Total apnea and respiratory distress time (RDT) throughout the test, expressed in minutes;Respiratory distress time index (RDTI)—duration of apnea and hypopnea episodes per test hour, expressed in minutes;Mean SpO_2_—mean saturation during the test;Minimal SpO_2_—minimal saturation during the test;T90—the percentage of test time where SpO_2_ was lower than 90%;Oxygen desaturation index (ODI)—the number of episodes of a decrease in SpO_2_ below 90% per hour of the test;Total oxygen desaturation time (ODT) throughout the test—the sum of the duration of episodes of decrease in SpO_2_ below 90%, expressed in minutes;Oxygen desaturation time index (ODTI)—desaturation time < 90% per test hour, expressed in minutes;Average heart rate (HR) throughout the study.

OSA was diagnosed based on polygraphy by the recommendations of AASM [24,25]. OSA was diagnosed in the case of AHI ≥ 15/h or AHI ≥ 5/h with accompanying clinical symptoms. The severity of OSA was assessed according to the following criteria:i.Mild disorders—AHI ≥ 5/h and <15/h, with accompanying clinical symptoms;ii.Moderate disorders—AHI ≥ 15/h and <30/h;iii.Severe disorders—AHI ≥ 30/h;

### 2.4. Measurement of Fetuin-A Level

Fetuin-A (Fet-A) was measured in a blood sample taken from the peripheral vein in the morning. Serum obtained after centrifugation of blood was stored at −20 °C until assays were performed. The Fet-A concentration was determined by the enzyme immunoassay ELISA method using the Human Fetuin-A ELISA Kit (BioVendor Laboratory Medicine, Brno, Czech Republic).

### 2.5. Statistical Analysis

The statistical analysis was performed using the STATISTICA 10 package (Statsoft Inc., Tulsa, OK, USA). Descriptive statistics included mean values and standard deviations. The compatibility of the distribution of the studied variable with normal distribution was assessed using the Shapiro–Wilk normality test. To assess the significance of differences, Student’s *t*-tests or variance analysis (ANOVA) in case of distributions not deviating from normal were performed. In cases of significant discrepancy with normal distribution, the nonparametric Mann–Whitney U test and Kruskal–Wallis ANOVA rang test were used. The evaluation of the variable’s dependencies was performed by determining the Pearson linear correlation coefficients for variables with normal distribution and the correlation coefficients of Spearman ranks in the opposite case. The statistical significance level of *p* < 0.05 was assumed.

## 3. Results

The study group of patients with OSA (OSA+) did not differ significantly from the control group (OSA−) in terms of age, sex, body weight, height, and BMI. The proportion of men was higher in the study group compared to the control group, but this was not a statistically significant difference. Patients in the study group scored significantly higher on the Epworth Sleepiness Scale compared to the control group (12.2 ± 5.1 vs. 7.9 ± 4.2; *p* = 0.0003). Both groups (OSA+ and OSA−) did not differ in terms of mean values of heart rate and systolic and diastolic pressure (Table 1).

As expected mean blood oxygen saturation was significantly lower and desaturation time significantly higher in OSA+ compared to OSA− group (Table 2).

In the PG-confirmed OSA (OSA+) group, the mean serum Fet-A concentration was 395.7 ± 62.8 μg/mL and was higher than in the control group, with PG-excluded OSA (OSA−) (378.3 ± 41 μg/mL), but this difference did not reach the assumed level of statistical significance. The severity of OSA did not affect Fet-A levels, which were 399.6 ± 80.0 μg/mL in group I (mild OSA), 381.1 ± 68.8 μg/mL in group II (moderate OSA), and 399.9 ± 56.9 μg/mL in group III (severe OSA) (*p* = NS). A significantly positive correlation of Fet-A with the pulse rate in PG was observed in both the study group (r = 0.335, *p* = 0.011) and the control group (r = 0.525, *p* = 0.005). No significant relationships of Fet-A concentration with other PG indicators studied were observed apart from its significant relationship with RDT in the control group (Table 3).

In both the OSA+ and OSA− groups, there was no dependence of Fet-A levels on systolic, diastolic, or mean arterial pressure. In the OSA− but not OSA+ group, a statistically significant (*p* = 0.02) positive correlation between Fet-A concentration and pulse pressure (r = 0.466) was demonstrated. The subjects OSA+ and OSA− were divided into three groups depending on the level of Fet-A: low (259.8–364.2 μg/mL), medium (364.6–405.8 μg/mL), and high (407.2–565.7 μg/mL). High Fet-A levels were most often found in the OSA+ group, whereas in the OSA− group, the most common Fet-A level was average (medium) (Figure 1).

The OSA+ group was divided into 3 subgroups depending on Fet-A concentrations: low (<364.2 μg/mL), medium (364.6–412.7 μg/mL), and high (>412.7 μg/mL). In the first subgroup, the mean HI value was the lowest, and in the third, it was the highest; these differences were statistically significant (*p* = 0.013) (Figure 2).

## 4. Discussion

OSA has both night and day symptoms. The first group includes loud snoring, restlessness, interrupted sleep, sudden awakening often with a feeling of fear, anxiety, shortness of breath, palpitations, increased blood pressure, increased sweating, nocturia, and increased physical activity at sleep. Day symptoms include difficulties with concentration, irritability, anxiety, depression, decreased libido, falling asleep while performing everyday activities, memory disorders, morning headaches, and dry mucous membranes of the upper respiratory tract [26]. OSA significantly affects the condition of the cardiovascular system in patients suffering from this syndrome. Repeated episodes of hypoxia leading to stimulation of the adrenergic system have a significant impact, for example, on the pathogenesis and clinical course of arterial hypertension, which is related to its clinical consequences [27,28]. One of them is the development of atherosclerosis, the pathogenesis of which may be influenced by Fet-A. This protein is supposed to prevent the development of organ calcification [29]. Hence, it can be assumed that its role in the pathogenesis of vascular diseases in the course of OSA may be significant. Fet-A is a glycoprotein of the α2-globulin fraction of the cystatin family, i.e., inhibitors of cysteine proteases; in adults, it is produced mainly in the liver [30]. Fet-A, previously known as Alpha 2-Heremans-Schmid glycoprotein (AHSG), discovered in humans in 1961 by Heremans, Schmid, and Burgi, is a protein composed of 349 amino acids and has a molecular weight of 46 kDa [31]. Fet-A is a hepato- and adipokine, as it is released into the blood from both hepatic cells and adipocytes [32]. Its concentration in blood serum is inversely proportional to the intensity of inflammation, and hence, it belongs to the so-called “negative” acute-phase proteins. In cases of acute bacterial infections, the concentration of Fet-A decreases, negatively correlating with CRP [33]. This is most likely due to the inhibitory effect of pro-inflammatory cytokines (interleukin 6 and interleukin 1β) on the concentration of Fet-A [34]. A positive correlation has been demonstrated between serum Fet-A concentration and insulin resistance, impaired glucose tolerance, and the risk of developing type 2 diabetes [35,36,37]. Among the multidirectional actions of Fet-A, there are also those limiting the calcification of vessels. These include preventing the accumulation of hydroxyapatite crystals [38], binding calcium and phosphate clusters [39], mediating the formation of calciprotein molecules [29], and inhibiting the formation and accumulation of calcium phosphates [40]. Fet-A level is related to cardiovascular mortality in dialysis patients. An increase of 0.1 g/l in the Fet-A level was associated with a significant reduction in all-cause mortality by 13% [41].

The associations of cardiovascular disease both with OSA, which increases CVD risk, and Fet-A, which prevents CVD, justify the question of their mutual interactions in patients with breathing disorders during sleep. It can be assumed that the hypothetical lack of protective effect of Fet-A in case of its secretion suppression in patients with OSA may be a factor increasing CVD risk. The results of our study do not indicate statistically significant differences in Fet-A levels in people diagnosed with OSA compared to the control group. Furthermore, the severity of OSA, measured by the number of apnea/hypopnea episodes or the parameters of reduced oxygen saturation, does not affect the level of Fet-A. However, high levels of Fet-A were found in subjects with OSA, and the hypopnea index differed significantly in groups with different Fet-A concentrations—it was highest among subjects with high levels. Fet-A levels did not correlate with systolic and diastolic blood pressure. The positive association of pulse pressure with Fet-A level observed in the control group did not occur in the sleep apnea group, which may suggest depression of the “Fet-A-dependent” protective mechanism in these patients. The hypothetical mechanism of such a phenomenon may lie in the known negative effect of inflammation on Fet-A levels. OSA is a condition conducive to the formation of systemic inflammation. In our study, Fet-A level positively correlated with the average pulse rate measured during polygraphy in both study and control groups. It has been shown that an increase in heart rate by 10 beats per minute was associated with an increase in the risk of cardiac death by at least 20%, and this increase in the risk is similar to the one observed with an increase in systolic blood pressure by 10 mm Hg. [42]. The increase in HR usually occurs in parallel with an elevation of blood pressure and metabolic disturbances (insulin resistance, dyslipidemia). Still, even after adjustment for the most important cardiovascular risk factors, HR remained an independent predictor of adverse events in the global population and a group of patients with cardio- and cerebrovascular diseases. [43] The obtained results, therefore, contradict the existence of a simple relationship between OSA and Fet-A metabolism. In our opinion, described Fet-A level changes were subtle but significant. To our best knowledge, our study is the only one to be performed in well-defined groups of patients with OSA without known metabolic or cardiovascular complications. The studies conducted so far have focused on patients with sleep apnea who had already been suffering from numerous diseases.

Results of previous investigations on Fet-A in OSA patients were contradictory. In the study by Barcelo et al. [44], significantly lower Fet-A level was found in the OSA+ group compared with the OSA− group. Fet-A level was significantly negatively correlated with apnea/hypopnea index. The tendency toward lower Fet-A values in patients with OSA and concomitant cardiovascular diseases was observed when compared with OSA patients without such diseases. There were no significant differences in Fet-A concentration in patients with and without diabetes. Based on the obtained results, the authors suggested that Fet-A may be one of the factors having a significant impact on the development of cardiovascular complications in patients with OSA. The correlation of Fet-A concentration with AHI demonstrated by them could indicate the inhibitory effect of sleep defragmentation and periodic hypoxia on the synthesis of Fet-A and its release into the blood. On the other hand, disturbed breathing during sleep is accompanied by chronic inflammation, in which, of course, one can expect down-regulation of the synthesis of proteins such as Fet-A. These conclusions could, according to the researchers, confirm the role of this protein as a useful predictor of clinical sequelae of OSA [44]. Lower levels of Fet-A were also observed in patients with OSA without hypertension compared to those without breathing disorders during sleep [45]. According to the authors of the study, a lower concentration of Fet-A in the OSA+ group is associated with subclinical atherosclerotic changes in the carotid arteries. Another study, similar to ours, showed no correlation between the concentration of Fet-A and the parameters of polygraphy in patients with OSA [46]. Fet-A levels did not differ depending on AHI and minimum night-time blood saturation or desaturation index. The degree of OSA severity was not found to have a significant effect on the level of Fet-A.

The incoherent results of the discussed studies may have several sources. Our study included subjects without comorbidities except for properly controlled arterial hypertension. In the study by Barceló et al. [44], both study and control groups included patients with diabetes and CVD, including patients with myocardial infarction in their medical history. There were more people with diabetes among subjects with OSA than in the control group. The presence of such serious comorbidities may have various effects on the concentration of Fet-A. Due to the different selection of patients, a direct comparison of the results of these two studies seems to be unjustified. Exploring the effect of OSA on serum Fet-A concentration in patients without comorbidities appears to be more proper. However, this is a big challenge, as most people with newly diagnosed OSA are already burdened with metabolic disorders, CVD, etc. The discrepancy in the results may also be influenced by the research methodology—in the work of Barcelo, full polygraphy performed was performed in contrast to our study, where simplified polygraphy was used. In the work of Akyuz et al. [45], lower Fet-A levels among OSA normotensive patients were found. However, there were significant changes in carotid artery stiffness and intima-media thickness, strongly suggesting atherosclerosis development in the examined group. Hence, despite lack of hypertension, co-morbidities among them clearly existed. Liu et al. [12] obtained the most similar results to those of our patients with OSA. This seems understandable, as the criteria for exclusion from the study that are closest to ours have been applied. These included a fasting glucose ≥ 126 mg/dL, history of diabetes or use of antidiabetic medications, CVD, kidney or liver diseases, or received treatment for OSA. As in our study, almost two-thirds of the cohort was male. Thus, there may also be a correlation with our results indicating the lack of a direct relationship of OSA with the Fet-A level. Due to the obtained results regarding Fet-A with metabolic syndrome, the authors believe that Fet-A affects the cardiovascular system, not indirectly, through the association with lipid disorders. Such compounds, also with the lipid profile in our study group, will be the subject of further research.

The strong point of our work is the selection of patients among whom there were no cases with comorbidities other than hypertension. Furthermore, the strict definition of the test and control group based on objective criteria (PG result), along with statistically insignificant differences in the distribution of anthropometric features, seems to be an asset to our study.

## 5. Limitations of the Study

The weakness of our work may be the methodology of the study based on simplified polygraphy and the relatively small size of the studied groups, which resulted from the scrupulous exclusion of people with metabolic and cardiovascular diseases from the study. That is why further studies are needed to establish the potential role of Fet-A as one of the cardiovascular risk markers in the complex context of oxidative stress and lipoprotein disturbances in patients with OSA [46,47,48]. The results of the salivary peptidome study seem to be promising in seeking novel biomarkers of early CVD development in patients with OSA [49].

## 6. Conclusions

Newly diagnosed OSA in patients without cardiovascular and metabolic comorbidities does not appear to directly affect Fet-A levels. The severity of OSA, the number of apnea and hypopnea episodes, as well as the severity of oxygen desaturation associated with apnea episodes did not correlate with the level of Fet-A. However, high Fet-A levels were more common in the group of patients with OSA, and the hypopnea index was significantly higher among subjects with the highest Fet-A levels. The level of Fet-A in OSA patients positively correlates with pulse rate, and it does not correlate with pulse pressure in this group unlike in the control group, where such a relationship exists. This suggests an impairment of the potential protective mechanism of increase in Fet-A levels in OSA patients preventing vessel wall calcification.

## Figures and Tables

**Figure 1 ijerph-19-06422-f001:**
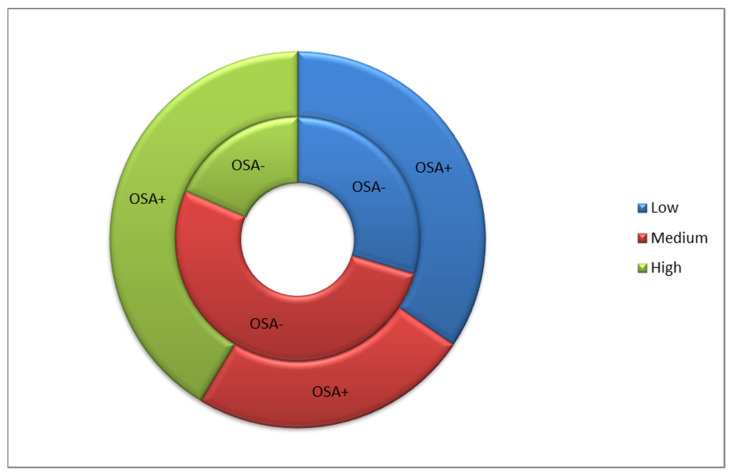
Fetuin-A level in OSA+ and OSA− groups.

**Figure 2 ijerph-19-06422-f002:**
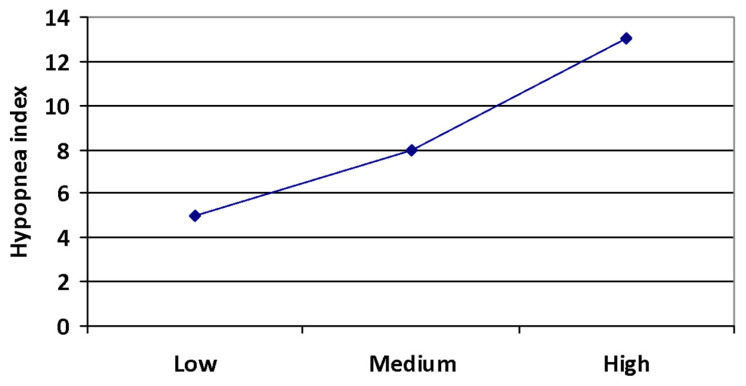
Hypopnea index (mean values) in the group with a low, medium, and high Fetuin-A level.

**Table 1 ijerph-19-06422-t001:** General characteristics of study (OSA+) and control (OSA−) groups.

Characteristics	Study Group (OSA+) (*n* = 58)	Control Group (OSA−) (*n* = 27)	*p*-Value
Age (years)	52.7 ± 9.5	48.2 ± 7.5	0.09
Male (%)	75.9%	66.7%	0.44
Female (%)	24.1%	33.3%	0.63
Height (cm)	173.6 ± 7.5	171.5 ± 11.3	0.12
Weight (kg)	95.8 ± 14.8	97 ± 15.1	0.81
BMI (kg/m^2^)	31.8 ± 4.3	33 ± 3.4	0.2
SBP (mmHg)	130.7 ± 12.5	126.5 ± 10.5	0.18
DBP (mmHg)	82.2 ± 8.3	80.6 ± 5.9	0.42
PP (mmHg)	48.5 ± 9.4	45.9 ± 9.5	0.17
MAP (mmHg)	98.3 ± 8.9	95.9 ± 6.3	0.19
Daytime sleepiness in ESS (points)	12.2 ± 5.1	7.9 ± 4.2	0.0003

BMI, body mass index; SBP, systolic blood pressure; DBP, diastolic blood pressure; PP, pulse pressure; MAP, mean arterial pressure; ESS, Epworth Sleepiness Scale.

**Table 2 ijerph-19-06422-t002:** Polygraphy results in the study (OSA+) and control (OSA−) groups.

Parameter	Study Group (OSA+) (*n* = 58)	Control Group (OSA−) (*n* = 27)	*p*-Value
AHI (h^−1^)	40.1 ± 21.6	2.6 ± 1.7	<0.001
AI (h^−1^)	29.7 ± 22.6	0.5 ± 0.6	<0.001
HI (h^−1^)	10.2 ± 8.2	2.1 ± 1.4	<0.001
RDT (min)	96.9 ± 66.5	5.5 ± 4.5	<0.001
RDTI (min/h)	17.6 ± 11.6	3.8 ± 11.7	<0.001
Mean SpO_2_ (%)	90.3 ± 3.4	92.6 ± 1.9	<0.05
Min. SpO_2_ (%)	72.8 ± 11	85.4 ± 3.9	<0.001
t90 (%)	19.7 ± 23	1.6 ± 3.5	<0.001
ODI (h^−1^)	43 ± 22.5	3.5 ± 1.9	<0.001
ODT (min)	101.4 ± 66.4	7.9 ± 4.4	<0.001
ODTI (min/h)	18.7 ± 11.6	1.3 ± 0.7	<0.001
Mean HR (min^−1^)	62.4 ± 8.1	61 ± 7	0.43

AHI, apnea/hypopnea index; AI, apnea index; HI, hypopnea index; RDT, respiratory distress time; RDTI, respiratory distress time index; SpO_2_, oxygen saturation, t90, SPO_2_ < 90% time; ODI, oxygen desaturation index; ODT, oxygen desaturation time; ODTI, oxygen desaturation time index; HR, heart rate.

**Table 3 ijerph-19-06422-t003:** Fetuin-A level and polygraphy results relationships in the study (OSA+) and control (OSA−) groups (significant results bolded).

Parameter	Study Group (OSA+) (*n* = 58)		Control Group (OSA−) (*n* = 27)	
Result	r	*p*-Value	r	*p*-Value
AHI (h^−1^)	−0.019	0.89	−0.293	0.14
AI (h^−1^)	−0.089	0.51	−0.056	0.78
HI (h^−1^)	0.194	0.14	−0.304	0.12
RDT (min)	−0.099	0.46	**−0.402**	**0.038**
RDTI (min/h)	−0.125	0.35	−0.215	0.28
Mean SpO_2_ (%)	−0.016	0.9	−0.131	0.51
Min. SpO_2_ (%)	−0.001	0.99	0.064	0.75
t90 (%)	0.028	0.83	0.208	0.3
ODI (h^−1^)	0.038	0.78	−0.073	0.72
ODT (min)	−0.002	0.99	−0.26	0.19
ODTI (min/h)	−0.027	0.84	−0.19	0.34
Mean HR (min^−1^)	**0.335**	**0.011**	**0.525**	**0.005**

AHI, apnea/hypopnea index; AI, apnea index; HI, hypopnea index; RDT, respiratory distress time; RDTI, respiratory distress time index; SpO_2_, oxygen saturation, t90, SPO_2_ < 90% time; ODI, oxygen desaturation index; ODT, oxygen desaturation time; ODTI, oxygen desaturation time index; HR, heart rate.

## Data Availability

Data supporting reported results are available upon request from the corresponding author.

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
