# Peer review of "Level of Serum Fetuin-A Correlates with Heart Rate in Obstructive Sleep Apnea Patients without Metabolic and Cardiovascular Comorbidities"

_ijerph, 2022, doi:10.3390/ijerph19116422_

Round 1

Reviewer 1 Report

The paper "Level of Serum Fetuin-A Correlates with Heart Rate in Obstructive Sleep Apnea Patients without Metabolic and Cardiovascular Comorbidities" by 
Elżbieta Reichert et al. provides an analysis of a group of patients with OSA and the impact of the serum fetuin A in OSA. There are some issues that can be solved.

  1. Page 2, line 56. Please define the properly controlled arterial hypertension.
  2. Page 2, line 77. Please describe how did you measure blood pressure. 
  3. Page 3, line111. Did you validate manually? Or automatically? Do you have an accredited sleep lab? With certified sleep physicians/technicians? 
  4. Page 3, line 118. How much does the measurement of Fet-A costs?
  5. Page 8, discussion. Do you have data from the impact of the medication, such as statins? 
  6. Page 9, discussion. A major limitation is the limited number of the patients. Can you comment more on that? 
  7. Reference 46. I did not find it  in the text. 
  8.  

Reviewer 2 Report

  1. OSA is commonly found in the adult population.- OSA affects only 2-9% adult population- please check this statement
  2. Please include the gender of participants and the numbers as OSA is more common in males compared to females. CVD develops at younger age in males. This may be a confounding factor
  3. Table 3- all “,” in numbers should be replaced by “.”.
  4. whereas in the OSA- group – the most common Fet-A level was average- means medium?
  5. The incoherent results of the discussed studies may have several sources.- please critically discuss the limitations for reference # 41, 42, 43
  6. Ref # 41 and 42- large text has been included, please include the takeaway message, no need to discuss the study.
  7. Please include the limitation of the study section.
  8. Please check yellow highlights

Reviewer 3 Report

Introduction

  • line 35, the criteria for defining obstructive sleep apnea were revised, defining an obstruction of at least 10 seconds, with total airway obstruction and thoraco-abdominal effort. please cite doi:10.5664/jcsm.6576.
  • line 40, Intermittent hypoxia and sleep fragmentation also lead to the production of oxygen free radicals, major neurodegenerative diseases, and poor cognitive performance. please cite doi:10.3390/bs11120180.
  • line 46, Patients with obstructive sleep apnea (OSA) are at increased risk for cardiovascular diseases (CVDs). Fetuin-A, a novel hepatokine, has been associated with the metabolic syndrome (MetS), insulin resistance, and type 2 diabetes mellitus, all of which are highly prevalent in patients with OSA and associated with increased CVD risk. please cite doi:10.1016/j.amjcard.2015.04.014.

Methods

  • please cite PSG guidelines
  • which type of PSG was performed?
  • how comorbidities where chosen?
  • specify committee code

In tables substitute p with p-value

Discussion

The salivary peptidome provides a promising approach to screening for novel biomarkers before further identification and may contribute to early diagnosis of CVD patients with OSA. please cite doi:10.1038/srep07046. 

Round 2

Reviewer 2 Report

None